# Association between Heated Tobacco Product Use during Pregnancy and Fetal Growth in Japan: A Nationwide Web-Based Survey

**DOI:** 10.3390/ijerph191811826

**Published:** 2022-09-19

**Authors:** Yoshihiko Hosokawa, Masayoshi Zaitsu, Sumiyo Okawa, Naho Morisaki, Ai Hori, Yukiko Nishihama, Shoji F. Nakayama, Takeo Fujiwara, Hiromi Hamada, Toyomi Satoh, Takahiro Tabuchi

**Affiliations:** 1Department of Obstetrics and Gynecology, Faculty of Medicine, University of Tsukuba, Tsukuba 305-8577, Japan; 2Center for Research of the Aging Workforce, University of Occupational and Environmental Health, Kitakyushu 807-8556, Japan; 3Institute for Global Health Policy Research, Bureau of International Health Cooperation, National Center for Global Health and Medicine, Tokyo 162-8655, Japan; 4Department of Social Medicine, National Center for Child Health and Development, Tokyo 157-8535, Japan; 5Department of Global Public Health, University of Tsukuba, Tsukuba 305-8577, Japan; 6Japan Environment and Children’s Study Programme Office, National Institute for Environmental Studies, Tsukuba 305-8506, Japan; 7Department of Global Health Promotion, Graduate School of Medical and Dental Sciences, Tokyo Medical and Dental University, Tokyo 113-8510, Japan; 8Cancer Control Center, Osaka International Cancer Institute, Osaka 541-8567, Japan

**Keywords:** heated tobacco products, small for gestational age, perinatal outcome, coronavirus disease 2019, smoking

## Abstract

Combustible cigarette smoking impacts fetal growth during pregnancy. However, the risk associated with heated tobacco products (HTPs) remains unclear. This nationwide cross-sectional study investigated whether HTP use during pregnancy is associated with small for gestational age (SGA) outcomes among 5647 post-delivery women with singleton pregnancies, which were divided into four groups: lifetime never-smokers, former smokers before pregnancy, and current smokers for each of the tobacco products during pregnancy (sole HTP and sole combustion smokers). Information on the prevalence of SGA, defined as birth weight and height below the 10th percentile, was retrieved from the Maternal and Child Health Handbooks of post-delivery women. Using logistic regression, the association between sole HTP smokers during pregnancy and SGA, adjusted for covariates, with lifetime never-smokers as reference, was investigated. The prevalence was: current sole HTP smokers during pregnancy, 1.8% (102/5647); and SGA, 2.9% (164/5647). Sole HTP smokers during pregnancy had a higher prevalence of SGA (5.9% [6/102] vs. 2.7% [111/4144]) with an adjusted odds ratio (OR) of 2.50 (95% confidence interval [CI], 1.03–6.05) than lifetime never-smokers. Among sole combustion smokers, the adjusted OR for SGA was 1.95 (95% CI, 0.81–4.67). In Japan, HTP smoking during pregnancy may be associated with an increased risk for SGA.

## 1. Introduction

Heated tobacco products (HTPs) have been available in Japan since 2014, and their market shares have been increasing worldwide, even during the coronavirus disease 2019 (COVID-19) pandemic [1,2,3,4,5]. Although some reports on the effects of HTPs on biomarkers in humans are available [6,7,8,9,10,11], information remains limited about whether HTP use increases adverse events in pregnant women and their infants [6,7,12]. The lack of evidence for HTP risk during pregnancy may have escalated HTP sales, particularly in the population who are at reproductive age [4,13].

We have previously reported perinatal complications associated with the use of HTPs [14], wherein an association between the use of HTPs and poor fetal growth was suggested. However, smoking status was classified as lifetime never-smokers or never-HTP-users, irrespective of pregnancy. Therefore, the specific adverse effects of HTP use during pregnancy on poor fetal growth remain unclear. In addition, the effect of relevant risk factors for poor fetal growth, including maternal pre-pregnancy body mass index (BMI) and weight gain during pregnancy, were not considered [15,16,17].

Small for gestational age (SGA) refers to an infant whose height and weight are below the 10th percentile and is diagnosed based on gestational age, sex of the infant, and parity [18,19,20]. SGA is associated with the risk of infant mortality and developing future-lifestyle-related disorders, such as hypertension and diabetes mellitus. SGA is also associated with maternal pre-pregnancy BMI and weight gain during pregnancy [16,17]. Therefore, identifying and exploring risks associated with SGA are crucial [15,21].

We hypothesized that the use of HTPs during pregnancy is associated with a higher risk of SGA because HTPs contain nicotine, which is one of the major causes of fetal growth restriction, including SGA [6,21]. Therefore, this study aimed to analyze the association between HTP use during pregnancy and the prevalence of SGA using data from a nationwide web-based survey in Japan.

## 2. Materials and Methods

### 2.1. Study Design, Data Setting, and Participants

This nationwide, cross-sectional study used data from an internet-based survey conducted as a part of the Japan COVID-19 and Society Internet Survey [14]. The data from the participants for each survey were retrieved from the pooled panels of an internet research agency [22]. First, a screening survey was conducted on 24 July 2021 to determine the eligibility of participants who had singleton births after July 2019, and 11,661 participants were determined to be eligible for this study. Next, the internet research agency distributed a questionnaire to all eligible post-delivery women through an e-mail with a link to a designated website. Data on current pregnancies were collected from 6256 women (response rate, 53.6%) between 28 July 2021 and 30 August 2021. First, we excluded 574 women who provided irrelevant or contradictory information and had inappropriate gestational weeks at delivery (<22 or >43 weeks). The definition was in accordance with the methods performed in previous studies by the same research group [14]. Second, to focus on the effect of HTPs, we excluded 35 women who smoked both HTPs and combustion cigarettes during pregnancy. Finally, 5647 women were eligible for our analysis. The prefectures where the selected post-delivery women lived were representative of all the prefectures in Japan (Appendix A).

As an incentive for study participation, the participants received credit points, called “Epoints”, which could be used for online shopping or could be converted to cash. Although the exact value of each Epoint was not disclosed per the internet survey agency’s rule, 1 Epoint was assumed to be equivalent to approximately 100 Japanese yen (approximately 1 US Dollar).

### 2.2. Definitions of Small for Gestational Age and Other Perinatal Outcomes

For the primary outcome of poor fetal growth, we chose SGA, which is defined as birth weight and height below the 10th percentile for each gestational age [18]. We applied the criteria of the Japanese Society of Pediatrics to determine SGA [23]. Except for self-reported parity (nulliparity or multiparity), the variables used to define SGA were based on answers that required the participants to refer to their Maternal and Child Health Handbook. All municipalities in Japan issue a handbook in which medical professionals record the health information of the mother and child, including clinical outcomes (e.g., blood pressure and birth weight) during pregnancy, to all pregnant women as part of the national maternal and child health policy. Mothers seldom lose their Maternal and Child Health Handbook (losing rate < 1%) [24].

Other perinatal outcomes, including the diagnosis of hypertensive disorders of pregnancy (HDP), were based on whether the participant had a history of hypertension or had a systolic blood pressure of ≥140 mmHg or a diastolic blood pressure of ≥90 mmHg during pregnancy [14,25], which was obtained by referring to the Maternal and Child Health Handbook. Similarly, the participants referred to their Maternal and Child Health Handbook to provide information on gestational weeks at delivery and the delivery method (vaginal or cesarean delivery). However, gestational diabetes status and an emergency cesarean section were self-reported as yes/no.

### 2.3. HTP and Cigarette Smoking

Participants were asked to report the number of combustible cigarettes and/or HTP smoking per day for the following three periods: before pregnancy, during the first trimester of pregnancy, and during the second and third trimester of pregnancy. Regarding combustible cigarettes (including hand-rolled or little cigarettes), we defined current combustible cigarette smoking as smoking any number of combustible cigarettes per day. Regarding HTPs, we asked the participants about the number of times they smoked any of the different types of HTPs (Ploom Tech, Ploom Tech plus, Ploom S, IQOS, glo, glo sens, and PULZE) per day at the time of the survey. We defined current HTP smoking as smoking any amount of HTPs per day.

The participants were divided into the following four groups according to their smoking status before pregnancy and during all the trimesters of pregnancy (first, second, and third trimesters): (1) lifetime never-smokers, who had never smoked any tobacco product in their lifetime; (2) former smokers, who quit smoking combustible cigarettes or HTPs before pregnancy; (3) current smokers of only HTP, who smoked solely HTPs during pregnancy; and (4) current smokers of only combustible cigarettes, who smoked solely combustible cigarettes during pregnancy. Dual smokers were already excluded while determining the selection of women eligible for the study.

For the sensitivity analysis to assess a dose–response association, sole HTP smokers were divided into two groups (<10 heat sticks/day and ≥10 heat sticks/day).

### 2.4. Other Covariates

Age (<25 years, 25–29 years, 30–34 years, 35–39 years, or ≥40 years), pre-pregnancy BMI (underweight < 18.5 kg/m^2^, normal 18.5–24.9 kg/m^2^, overweight 25.0–29.9 kg/m^2^, or obese ≥ 30 kg/m^2^), and maternal gestational weight gain (insufficient, appropriate, or excessive) were included as basic confounding variables. As the appropriate value for gestational weight gain depends on the pre-pregnancy BMI, we considered both pre-pregnancy BMI and gestational weight gain during pregnancy [16,17]. Thus, we classified the participants into 11 categories as an anthropometric parameter of mothers to predict SGA (Appendix A) [17,26]. As socioeconomic and environmental variables [14], we included household income (<5 million JPY [approximately 50,000 USD], 5 to <8 million JPY, ≥8 million JPY, or unknown), occupation (manager or others), educational attainment (≤12 years [high school] or ≥13 years [college or university]), prefectures where the women resided (prefectures with >5 million population and those with <5 million population), and date of delivery (July 2019–February 2020, March 2020–August 2020, September 2020–February 2021, or March 2021–August 2021). We also included self-reported comorbidities (chronic kidney disease and/or autoimmune disease) that are the risk factors for SGA [27].

### 2.5. Statistical Analyses

To assess the risk of SGA, we estimated the odds ratios (ORs) and 95% confidence intervals (CIs) of each smoking category using logistic regression analysis. The reference group was lifetime never-smokers. We adjusted for basic confounding variables (maternal age, pre-pregnancy BMI, and gestational weight gain) in model 1. In model 2, ORs were additionally adjusted for socioeconomic and environmental variables (household income, occupation, educational attainment, the prefecture where the women resided, and date of delivery). In model 3, ORs were fully adjusted for clinical variables of HDP and comorbidities (chronic kidney disease and autoimmune disease), which are the known risk factors for SGA [27]. In the sensitivity analysis, we estimated the risk of SGA in sole HTP smokers who smoked <10 heat sticks per day and sole HTP smokers who smoked ≥10 heat sticks per day compared with lifetime never-smokers in the same analysis.

All statistical analyses were performed using IBM SPSS 26.0 for Windows software (IBM., Armonk, NY, USA). A two-sided *p*-value of < 0.05 was considered statistically significant.

### 2.6. Ethics Approval

This study was approved by the Bioethics Review Committee of Osaka International Cancer Institute, Japan (approval number 20084-2). All procedures in this study were performed in accordance with the ethical guidelines for medical and health research involving human participants enforced by the Japanese government’s Ministry of Health, Labour, and Welfare and the 1964 Helsinki Declaration and its later amendments. Informed consent was obtained electronically, and all participants were informed of their right to withdraw from the study.

## 3. Results

Among the 5647 study participants, the prevalence of current HTP only smokers during pregnancy was 1.8% (*n* = 102, Figure 1). The background characteristics of pregnant women differed across smoking categories (Appendix A). The prevalence of SGA among current sole HTP smokers and lifetime never-smokers was 5.9% (6/102) and 2.7% (111/4144), respectively (Table 1). Additionally, the prevalence of pre-term births was 8.8% among current sole HTP smokers and 5.5% among lifetime never-smokers (Table 1).

In the logistic regression analysis, the odds for SGA prevalence were significantly increased in current sole HTP smokers; the OR was 2.70 (95% Cl, 1.14–6.40) in model 1 (Table 2, Figure 2, and Appendix A). After adjusting for socioeconomic and environmental variables (model 2) and clinical variables (model 3), the elevated odds among current sole HTP smokers were attenuated slightly; albeit, the significance remained (Table 2, Figure 2). The ORs in models 2 and 3 were 2.49 (95% CI, 1.04–5.98) and 2.51 (95% CI, 1.04–6.07), respectively. For current sole combustion smokers, the odds for SGA were 1.95 (95% CI, 0.81–4.67) in model 3 (Table 2). Among former smokers, the magnitude of OR for SGA was likely smaller than that among current smokers, and the odds were not significant (Table 2).

In addition, maternal age and HDP were associated with SGA prevalence (model 3, Appendix A). Regarding the maternal anthropometric parameter, low weight gain during pregnancy was associated with a higher risk of SGA among pregnant women who were underweight and normal weight before pregnancy (Appendix A). In contrast, among pregnant women with a pre-pregnancy BMI of ≥25 kg/m^2^, weight gain below the threshold during pregnancy was not associated with an increased risk of SGA.

In a sensitivity analysis among current sole HTP smokers, the odds for SGA prevalence in the highest category (i.e., ≥10 heat sticks per day) were most pronounced (Figure 2 and Appendix A), thereby suggesting a potential dose–response tendency.

## 4. Discussion

In this nationwide, cross-sectional study, we demonstrated that HTP use during pregnancy is associated with SGA. Compared with pregnant women who never smoked during their lifetime, those who used HTPs during pregnancy had a 2.5-times higher risk of SGA. Our results indicate that HTP use during pregnancy has an unfavorable impact on infants.

Despite insufficient research on the acute effects of HTPs, the rapid increase of HTP use globally is a serious public health concern, particularly among the population at reproductive age. For instance, in Japan, where the global market share of HTPs is the highest (~>90%), the current prevalence of smoking HTPs is approximately 16% among the younger population, aged 20 to 29.2 years. The use of HTPs has been reported to cause changes in biomarkers and other parameters [6,7,8,9,10,11], suggesting that HTPs-related oxidative stress/systemic inflammation may result in unfavorable perinatal outcomes for mothers and/or infants. In this study, we demonstrated that HTP smoking during pregnancy is associated with SGA. Additionally, a sensitivity analysis showed a dose–response association between SGA and the amount of HTPs. As previously reported [21,28], combustible cigarette smoking is an established risk factor for poor fetal growth. Given that HTPs are “tobacco” products and the odds of former smokers who quit any type of tobacco smoking before pregnancy were not elevated in our study, our results are plausible. Yet, biological mechanisms underlying the association between HTP smoking and SGA remain unclear.

Interestingly, our categories for pre-pregnancy BMI, further classified by weight gain during pregnancy, which was significantly associated with SGA in this study, might have better precision in assessing the body size and weight change of pregnant women. Therefore, further studies for a weight gain index during pregnancy for each pre-pregnant BMI category are warranted.

The limitations of our study should be noted. First, we could not explain causality because the study used a cross-sectional design with retrospective reporting. People who prefer new tobacco products, such as HTPs, might be engaging in unadjusted/unobserved SGA-related risk behaviors before smoking HTPs, which were not adjusted as confounders in our analysis. Thus, it cannot be denied that the use of HTPs may be a mere mediator for the association between risky lifestyle behaviors (e.g., alcohol or drug consumption) and SGA. However, sensitivities for our main findings were fair to consider unadjusted/unobserved confounding factors. For example, the E-value for the fully adjusted OR in sole HTP smokers for SGA was 4.44. The E-value is defined as the minimum strength of association on the risk ratio scale that an unmeasured confounder would need to have with both the treatment and outcome to fully explain a specific treatment–outcome association, based on the measured covariates [29]. Second, smoking status was self-reported, and we did not confirm smoking status with biomarkers such as blood or urine cotinine [30,31]. Additionally, we did not assess the combined effects of combustible smoking and HTP smoking on fetal growth due to the small sample size in this category. As pregnant women generally tend to conceal their smoking habits during pregnancy, some current combustible cigarette and HTP smokers might have possibly been misclassified into lifetime never- or former smokers [31]; this might have introduced social desirability bias. Therefore, our estimates would have been biased toward the null. Third, the information for SGA was not directly retrieved from medical charts but was self-reported by referring to the Maternal and Child Health Handbook. Although the existence of input error cannot be denied, this could occur at random. Fourth, the small number of observed outcomes might have introduced sparse data bias (e.g., the OR of HTP smokers for SGA compared with lifetime never-smokers was 2.19, with a 95% penalized profile-likelihood posterior CI of 1.00–4.34 in a univariate penalized logistic regression) [32]. Despite these limitations, the strength of this study is that we distinguished pregnant women who smoked solely HTPs during pregnancy from those who were combustible cigarette smokers. In addition, to the best of our knowledge, this is the first and largest nationwide study showing neonatal adverse effects with the use of HTPs during pregnancy. Therefore, further larger studies with non-sparse data are warranted to elucidate the risk of maternal HTP smoking during pregnancy for their infants.

## 5. Conclusions

HTP smoking during pregnancy may be associated with an increased risk for SGA in Japan. SGA is a serious condition associated with the development of post-natal diseases and is also a risk factor for infant mortality [15,21]. Currently, many young people, particularly those at reproductive age, believe that HTPs are less harmful than combustible cigarettes [13]. However, HTP smoking is likely to have the same perinatal risks as conventional combustible cigarettes. Hence, public awareness of HTP-associated perinatal risks and preconception care at schools should be further encouraged. Women at their reproductive age are advised to quit HTP smoking for their health and that of their future infants.

## Figures and Tables

**Figure 1 ijerph-19-11826-f001:**
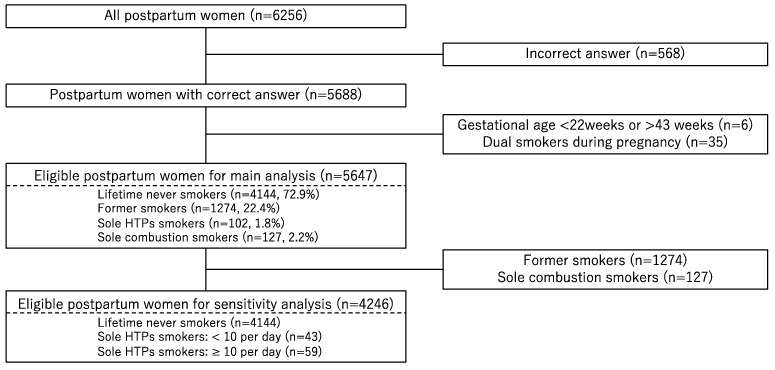
Flowchart of post-delivery women recruitment.

**Figure 2 ijerph-19-11826-f002:**
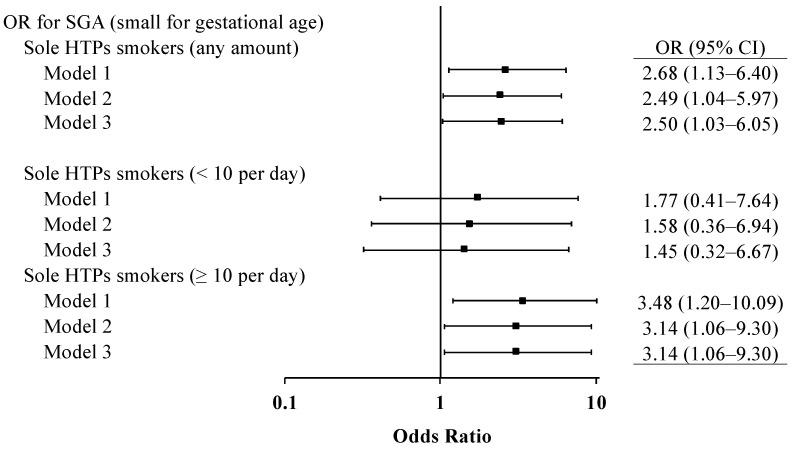
Odds ratios of current sole HTP smokers (any amount, <10 per day, and ≥10 per day) for SGA compared with never-smokers. CI, confidence interval; HTPs, heated tobacco products; OR, odds ratio; SGA, small for gestational age.

**Table 1 ijerph-19-11826-t001:** Perinatal outcomes across smoking categories among 5647 post-delivery women with singleton pregnancy.

Perinatal Outcomes	Mean ± SD or N (%)			
	Lifetime Never-Smokers *n* = 4144	Former Smokers *n* = 1274	Sole HTP Smokers *n* = 102	Sole Combustion Smokers *n* = 127
Gestational weeks < 37	236 (5.7)	72 (5.7)	9 (8.8)	10 (7.9)
Male sex	2067 (49.9)	608 (47.7)	56 (54.9)	65 (51.2)
Delivery mode				
Vaginal	3344 (80.7)	1021 (80.1)	86 (84.3)	105 (82.7)
Scheduled cesarean	359 (8.7)	108 (8.5)	9 (8.8)	10 (7.9)
Emergency cesarean	441 (10.6)	145 (11.4)	7 (6.9)	12 (9.4)
HDP	277 (6.7)	111 (8.7)	8 (7.8)	14 (11.0)
GDM	226 (5.5)	75 (5.9)	10 (9.8)	8 (6.3)
Birth weight (g)	3008 ± 416	3011 ±4 31	2994 ± 449	2948 ± 376
SGA	111 (2.7)	40 (3.1)	6 (5.9)	6 (4.7)

GDM, gestational diabetes mellitus; HDP, hypertensive disorders of pregnancy; HTPs, heated tobacco products; SD, standard deviation; SGA, small for gestational age.

**Table 2 ijerph-19-11826-t002:** Odds ratios of each smoking category for the prevalence of small for gestational age.

Smoking Status	Odds Ratio (95% Confidence Interval)			
	Univariable	Model 1 ^a^	Model 2 ^b^	Model 3 ^c^
Lifetime never-smokers	Reference	Reference	Reference	Reference
Former smokers	1.18 (0.82–1.70)	1.29 (0.89–1.87)	1.24 (0.85–1.81)	1.23 (0.84–1.79)
Sole HTP smokers	2.27 (0.97–5.29)	2.68 (1.13–6.40)	2.49 (1.04–5.97)	2.50 (1.03–6.05)
Sole combustion smokers	1.80 (0.78–4.18)	2.13 (0.90–5.03)	1.96 (0.82–4.69)	1.95 (0.81–4.67)

HTP, heated tobacco product. ^a^ Odds ratios were estimated with logistic regression, adjusted for maternal age, pre-pregnancy body mass index, and gestational weight gain. ^b^ Additional adjustment for household income, occupation, education attainment, living in a prefecture with >5 million population, and date of delivery. ^c^ Fully adjusted for hypertensive disorders of pregnancy, chronic kidney disease, and autoimmune diseases.

## Data Availability

The data that support the findings of this study are available from the corresponding author upon reasonable request. However, restrictions apply to the availability of these data because data associated with personal identification cannot be shared. If any person wishes to verify our data, they are most welcome to contact the corresponding author.

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
