# Peer review of "Association between Heated Tobacco Product Use during Pregnancy and Fetal Growth in Japan: A Nationwide Web-Based Survey"

_ijerph, 2022, doi:10.3390/ijerph191811826_

Round 1
Reviewer 1 Report
This is a good paper investigating a newly emerging maternal and infant health issue.
I have a couple of queries and minor but important points to make.
The authors state they have excluded Dual Smokers from the analysis. I wonder if this is a missed opportunity to also assess the combined effects of combustible and HCT nicotine delivery on fetal growth, and compare that to combustible tobacco only, HCT only, and never and former smokers. Unless I’ve misunderstood the study cohort, Dual Smokers could have been included as a separate category for analysis – though I accept there were only 35 Dual Smokers in the sample.
Did JACSIS contain variables around other risky behaviours like alcohol or drug consumption that may also impact fetal growth? If so, could these represent additional confounders that might be controlled for in your model?
In your discussion, you mention that you feel recall bias was not present because little is known about the adverse effects of HTP of fetal growth. However, I think it is more an issue of reduced social desirability effects than recall bias, given low public and scientific knowledge about HTP use during pregnancy.
Limitations section was good overall.
Reviewer 2 Report
This nationwide cross-sectional study on 5647 postdelivery women in Japan examined the association between heated tobacco product (HTP) use during pregnancy and small for gestational age (SGA). The study found a higher odds of SGA in current HTP users during pregnancy (5.9% vs 2.7%; adjusted OR 2.50) than in lifetime never smokers. The results remained strong and significant after adjusting for other prognostic factors of SGA, including gestational weight gain (GWG), maternal age, and pre-pregnancy BMI. The E-value to nullify the association was 4.4, which was high and indicates the robustness of the results to confounding factors.
The health consequences of HTP use have remained unclear. This study adds to the literature by showing that HTP use during pregnancy could lead to SGA. However, I have the following concern and suggestions:
1. My main concern is the small number of outcome events by exposure categories (only 6 cases of SGA in each of the current HTP and cigarette users), which may lead to sparse data bias. This issue is even more concerning in the subgroup analyses shown in figure 2. Further statistical manipulation may be warranted to address this concern (see https://www.bmj.com/content/352/bmj.i1981).
2. Introduction: please include the potential biological mechanisms that underly the effect of HTP use on SGA for justifying the need of examining their association.
3. Suggest briefly introducing known pregnancy factors (e.g., GWG, GDM, pre-pregnancy BMI) associated with SGA, which help explains the choices of confounding variables in the models
4. As GWG is highly prognostic of SGA, more information about its measurement should be given. For instance, how “insufficient”, “appropriate” and “excessive” GWG were defined?
5. Results: I could not find the supplementary tables S1 to S3 and figure S1 in the manuscript file and thus could not comment on them.
6. Suggest briefly describing the meaning and utility of E-value, which has remained unknown to many researchers.
Round 2
Reviewer 2 Report
The authors adequately addressed my concerns. I have no further comments